# Analysis of Runoff According to Land-Use Change in the Upper Hutuo River Basin

Bin Liu [1,2], Jie Yang [1,2], Jinxia Sha [3,*], Yun Luo [1,2], Xian Zhao [1,2] and Ruiting Liu [1,2]

1   School of Water Conservancy and Hydroelectric Power, Hebei University of Engineering, Handan 056021, China
2   Hebei Key Laboratory of Intelligent Water Conservancy, Handan 056001, China
3   School of Earth Science and Engineering, Hebei University of Engineering, Handan 056021, China
*   Correspondence: shajinxia@163.com; Tel.: +86-310-312-3702

**Abstract:** Land use affects regional hydrological processes. The alteration of regional distributions of vegetation, crop types, and land-use patterns for construction has a significant impact on the runoff process and influences the water cycle in watersheds. Studies on runoff variations in the Hutuo River Basin have concentrated on climate change and the effect of human activities without adequate attention paid to land-use changes. In order to investigate the response of runoff to land-use changes in the upper Hutuo River Basin, a soil and water assessment model was used in this study to compare and analyze the changes in runoff under five land-use scenarios from 1980–2020. The results show that the area of farmland, forest land, and grassland in the watershed gradually decreased from 1980 to 2020, with a total decrease of 3.1%, while the area of urban construction land increased rapidly by 1.5 times. Corresponding with the trend of land-use change, the differences between the simulated and natural values for regional flood peak and annual runoff increased with time, which is in line with the changing land-use trends. From 1960–2020, the differences between the simulated and natural values for the flood peaks of the five land-use scenarios were −16.8, −6.7, −3.5, 4.6, and 9.3%, respectively, and the errors between the simulated and natural values for annual runoff were −6.7, −4.4, −2.0, −2.6, and 10.8%, respectively. Overall, the increase in urban construction land and decrease in farming, forest area, and grassland has caused the regional flood peak and annual runoff volume to increase in the upper Hutuo River Basin.

**Keywords:** land use; runoff change; soil and water assessment tool model; upper Hutuo River Basin; construction impact; regional flood peak

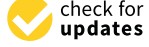



## 1. Introduction

The impact of land use on changes in the hydrological situation refers to the effects of changes in forests, grasslands, agriculture, industrialization, and urbanization on the various systems of the water cycle of the watershed. Land-use changes impact various water cycle elements because they alter the surface conditions of the watershed, which, in turn, affect runoff generation, confluence, and evapotranspiration [1]. The formation of flow processes at the outlet section of the basin depends on the surface conditions, wherein meteorological processes, such as rainfall and evapotranspiration, are similar [2]. The process of rapid human socioeconomic development in terms of the construction of engineering facilities and changes in farmland area and spatial planning lead to changes in land use, which, in turn, will affect the hydrological evolution of the watershed [3–6]. Therefore, it is important to study the impact of watershed land-use changes on runoff processes.

As early as 1851, Mulvanyt [7] established a quantitative link between flood flow—a distinctive hydrological variable—watershed area, and surface characteristic coefficients, thereby pioneering the investigation of the quantitative link between hydrological processes and surface conditions in watersheds. Previous researchers mostly used experimental watershed methods, such as the control watershed experimental treatment method [8,9]

and the parallel comparison and observation method of similar watersheds [10,11], to study the effects of land-use types on runoff. The experimental method can reveal the hydrological effects of land use/vegetation cover to some extent, but it cannot be carried out effectively because of limitations, such as a long experimental period, difficult operation, and large differences in the natural conditions of the watersheds. In recent years, scholars have used statistical analysis and modeling methods to explore the effects of land-use change on runoff. In terms of statistical analysis, Wang et al. [12] explored the impact of one aspect of land use, reforestation measures, on runoff based on a new attribution method in the Budyko framework; reforestation led to a significant reduction in annual runoff in the Loess Plateau region. Zhang et al. [13] established a vegetation–parameter–runoff analysis framework based on the elastic coefficient approach, starting from another aspect of land use, the vegetation cover index, and discovered that changes in the normalized vegetation index significantly influenced the runoff of the Huang-Huai-Hai River Basin. No longer restricted to a single factor in land use but instead based on regression analysis and Budyko hypothesis, Lei et al. [14] revealed the reasons for the sharp decrease in runoff in the Sanchuan River Basin in Western Jin from 1980 to the present, based on the overall trend of land use in the basin. Khoie et al. [15] compared land use and climate change as the two main factors influencing runoff, using simple differential method (SDM) and climate elasticity method (CEM) estimations, showing that land-use changes contribute more than 60% to runoff changes in all the sub-basins of the Gorganroud basin in Iran. In terms of hydrological simulations, Cuo et al. [16] used an improved variable infiltration capacity model to demonstrate that land-use and land-cover changes have a greater impact on runoff than climate change in areas with relatively intense human activity in the upper reaches of the Yellow River. Liu et al. [17] used the distributed hydrological soil vegetation model (DHSVM) to further demonstrate that climate change and land use change act to different degrees in the different sub-basins of the same watershed. Instead of looking at the overall impact of land use and climate change on runoff, Khorn et al. [18] explored the effect of change in an area on runoff for different land-use types based on the SWAT model, with land-use changes in the study area mainly being the conversion of forest land to agricultural land, leading to an increase in the multiyear average surface runoff in the watershed. Similar to Khorn [18], Zhang et al. [19] also used the SWAT model for quantitative analysis and concluded that an increase in forested land area, as well as a decrease in cultivated land area, would lead to a decrease in runoff. For dry season flow, Tasgara et al. [20] used a maximum likelihood classifier to classify Landsat images and generate land-use maps to quantify the impact of land-use change on watershed runoff. They discovered that the decrease in dry season flow was mainly due to a decrease in forest and grassland area and an increase in agricultural and built-up land area. In addition to studying the impact of existing land use on runoff, Tang [21], Ahmadi [22], and Mfwango [23] also used the hydrological model method to predict the impact of future land use on runoff. Tang et al. [21] used the future land-use simulation model for land-use projections in the Miluo River Basin for 2035 and assessed the changes in the hydrological response based on different land-use scenarios using the soil and water assessment tool (SWAT) model. Ahmadi et al. [22] applied the cellular automata–Markov chain model to generate land-use maps for 1996, 2008, 2018, and 2033 to analyze the impact of land-cover change on future runoff. Mfwango et al. [23] explored the impact on runoff by predicting land-use maps for 2040 and 2070 using Land Change Modeler (LCM) based on existing land-use patterns. In order to explore what is more applicable to hydrological research—the statistical analysis method or the modeling method—Liu et al. [24] concluded that statistical analysis could only reveal the overall basin variability over long timescales, and hydrological models could reflect temporal and spatial differences in the variability of hydrological processes.

Over the years, many scholars have used different methods to analyze the impact of land use on runoff from different levels and perspectives. In order to investigate the impact

of the spatial and temporal changes in land use on runoff in the Hutuo River Basin, this paper uses the SWAT model and the hydrological modeling approach.

The Hutuo River is a major river in the Haihe River Basin and supports more than 10 million people. The basin has the Huangbizhuang and Gangnan Reservoirs, which are responsible for delivering water to urban and rural populations, industry, and agriculture. Over the last two decades, studies on the runoff variations of the Hutuo River Basin have concentrated on climate change and the effect of human activities, with inadequate attention paid to land-use changes. Zhao et al. [25] explored the response of runoff in the Hutuo River Mountains in two more general directions: climate change and the impact of human activities. The results showed that the decrease in water resources in the Hutuo River mountain area from 1980 to 2010 was influenced by climate change and human activities and that human activities were the dominant factor. Wang et al. [26] also conducted a proportional analysis of the degree of influence on runoff from the Hutuo River in terms of both climate change and human activities. They found that 26.7% of the decrease in runoff in the Hutuo River basin from 1980 to 2013 was attributed to climate change and 73.3% to human activities. While Miao et al. [27] further quantified the contribution of climate change and human activities to runoff changes in the Hutuo River on this basis, concluding that the impact of human activities on runoff is increasing, they still did not further mention the impact of land use on runoff in the Hutuo River. Xi et al. [28] analyzed land-use change in the Hutuo River Basin from 2000–2015 and projected the land-use patterns in the study area in 2025 but did not explore the relationship between land-use change and runoff.

In this study, the upper reaches of the Hutuo River Basin were studied to clarify the impact of land-use changes on runoff and provide a rationale for the development and utilization of regional water resources, sustainable socioeconomic development, and the implementation of adaptive countermeasures for the protection of water resources in changing environments.

## 2. Overview of Research Area

The Hutuo River runs from Fanshi, Shanxi Province, through Shanxi to Hebei Provinces. The Hutuo River has a total length, arial extent, and elevation of 605 km, 25,168 km$^2$ ($112°13'114°19'$ E, $37°17'39°27'$ N), and 66–3072 m, respectively. The basin is located in a temperate semihumid semiarid monsoon climatic zone with an average annual temperature of 9.71 °C. The basin's multiyear average precipitation (1960–2020) is 472.72 mm, and the distribution across the year is quite irregular, with 83.19% of the annual precipitation concentrated during May–September. The three land types that make up the majority of the watershed, such as forest land, cropland, and grassland, account for nearly 93.08% of its total area, with grassland and forest land making up >67.53% of that total. The basin above the Huangbizhuang Reservoir is the object of this study. The study area is 23,932 km$^2$, accounting for 95.09% of the entire Hutuo River Basin. There are two large reservoirs in the study area. The Gangnan Reservoir has a capacity of 1.57 billion m$^3$ and a control basin area of 15,900 km$^2$. The Gangnan Reservoir is a large-scale water conservation project that comprehensively supports flood control, irrigation, power generation, and fish culturing. The Huangbizhuang Reservoir, located 28 km downstream of the Gangnan Reservoir, has a total storage capacity of 1.21 billion m$^3$ and a control basin area of 23,400 km$^2$. The Huangbizhuang Reservoir is mainly utilized for flood management, urban water supply, agriculture, power generation, and other diverse applications. The study area is illustrated in Figure 1.

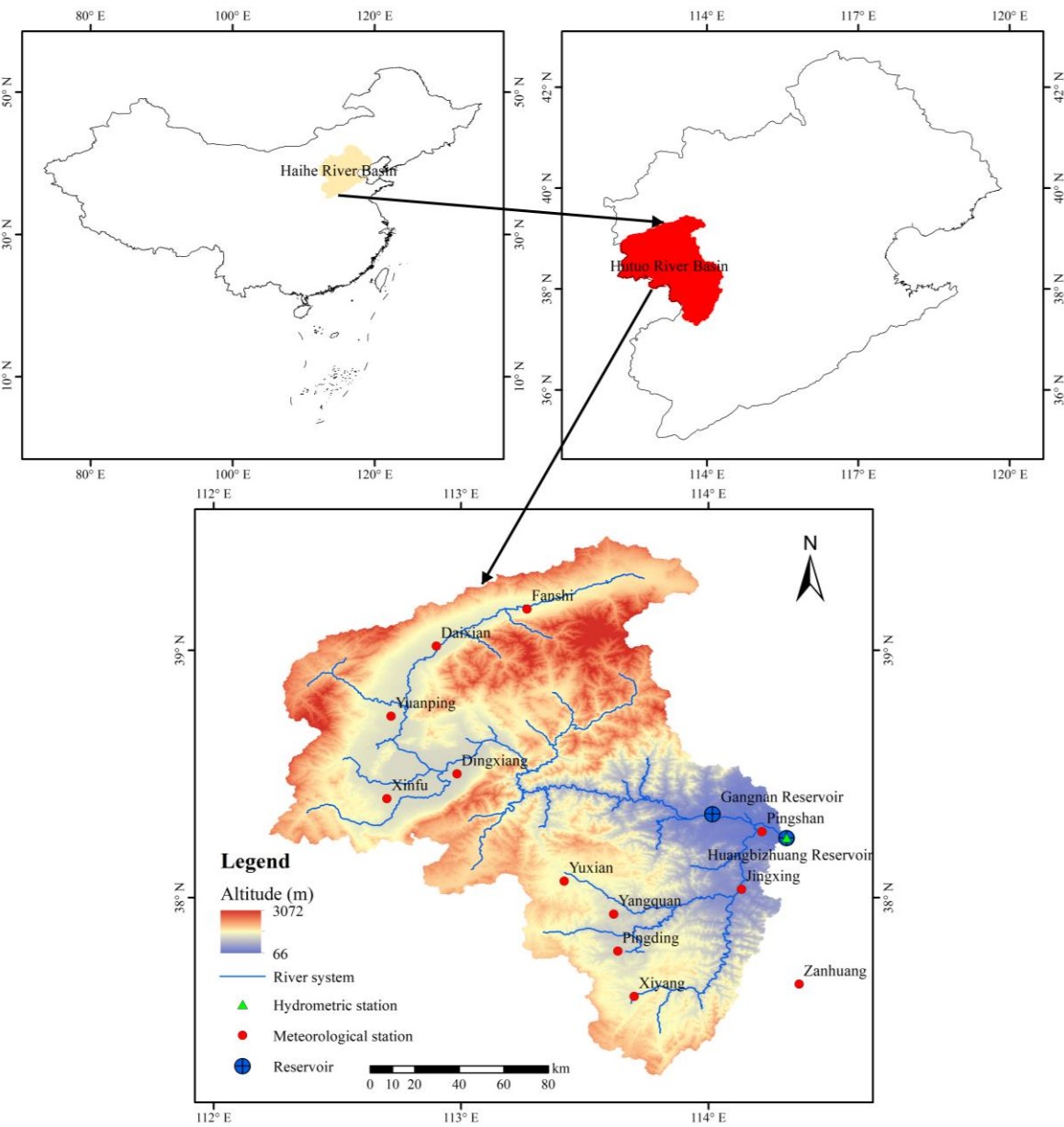

**Figure 1.** Geographical map of the upper Hutuo River Basin.

## 3. Methodology

### 3.1. SWAT Model

The SWAT model is a semidistributed model based on physical process development created by the US Department of Agriculture's Agricultural Research Center in 1994 [29,30]. The distributed hydrological model, as opposed to the lumped hydrological model, considers climatic factors and surface conditions and can accurately simulate the impact of changes in climatic conditions, land use, soil types, management measures, and other conditions on the processes of the basin's hydrological cycle [31–33]. The SWAT model is extensively used to quantitatively examine the influence of climate change and human activity on runoff because it can simulate continuous time series and accurately express hydrological variables and their evolution.

This study classified the upper reaches of the Hutuo River Basin into 64 sub-basins based on factors such as the natural flow direction of the river, the control area of the hydrological station, and the location of the reservoir to facilitate simulation of the spatiotemporal variability of the surface and climatic factors.

### 3.2. Data Preparation

The types and quantity of data required for the implementation of the SWAT model are substantial. Meteorological data, runoff data, digital elevation maps, land-use maps, soil maps, and reservoir characteristics (in the research region) were necessary for this study. Table 1 displays the resolution, source, and length of the time series of each dataset.

**Table 1.** Characteristics of data used for the study.

| Number | Data Type | Resolution | Year | Source |
|---|---|---|---|---|
| 1 | Meteorological data | Daily | 1960–2020 | China Meteorological Date Service Center |
| 2 | Runoff data | Daily | 1960–2020 | Hydrological Yearbook of the Hai River Basin |
| 3 | Digital Elevation Map | 30 m | 2020 | Geospatial Data Cloud |
| 4 | Land-use map | 1 km | 1980, 1990, 2000, 2010, and 2020 | Resource and Environmental Science and Data Center, Chinese Academy of Sciences |
| 5 | Soil map | 1:1 million | - | World Soil Database |
| 6 | Monthly outflow of Huangbizhuang Reservoir | Daily | 1960–2020 | Hydrological Yearbook of the Hai River Basin |

The land-use data used in this study were obtained by interpreting remote sensing images for the five years 1980, 1990, 2000, 2010, and 2020. To meet the requirements of the SWAT model, the land-use data of the watershed were reclassified into six categories by using the reclassification tool in Arcgis, such as arable land, forest land, grassland, water, urban construction land, and unused land. The spatial distribution of land use in each decade is shown in Figure 2.

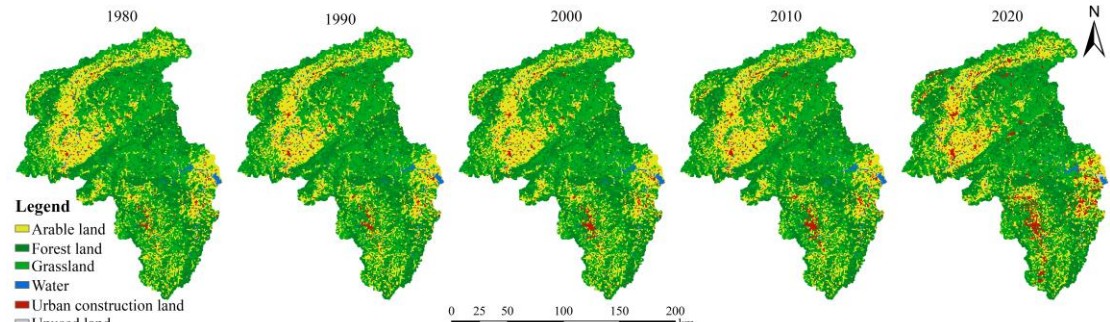

**Figure 2.** Spatial pattern of land-use changes in the upper Hutuo River Basin from 1980 to 2020.

The spatial distribution map of soil in the upper reaches of the Hutuo River Basin was extracted, and the soil was reclassified according to the Chinese soil group in the Harmonized World Soil Database. After reclassification, the soil parameters were calculated using Soil–Plant–Atmosphere–Water software using a statistical analysis approach.

Data from 12 national meteorological stations in Daixian, Fanshi, Yuanping, Xinfu, Dingxiang, Yuxian, Pingding, Jingxing, Pingshan, Yangquan, Xiyang, and Zanhuang were used. Data on daily precipitation, wind speed, average temperature, maximum temperature, lowest temperature, relative humidity, and sunshine duration were utilized for the duration of 1960–2020. SWAT Weather database was used to calculate and build the weather generator, and a meteorological data index table was created.

The measured runoff of a hydrological station was restored to natural runoff because of water abstraction along the river upstream of the Huangbizhuang Reservoir to accurately replicate the effect of land use on runoff.

### 3.3. Calibration and Validation

As the SWAT model uses numerous parameters, automatic parameter adjustment was used to find optimal parameters in a shorter time compared to manual parameter adjustment. Consequently, an automatic calibration program with an appropriate objective function was preferred. SWAT-CUP links optimization algorithms, such as particle swarm optimization, sequential uncertainty fitting algorithm (SUFI−2), parameter solution, and Markov chain Monte Carlo method with SWAT, to perform sensitivity analysis, calibration, verification, and uncertainty analysis [34,35]. The SUFI−2 algorithm developed by Abbaspour et al. used the Latin hypercube random sampling method to obtain simulation parameter values, which were then substituted into the SWAT model to calculate the objective function values, and the optimal parameters of the model were finally determined through parameter sensitivity analysis and multiple iterations of the parameters [36,37]. The algorithm is widely used because it can combine the subjective and cognitive aspects of the analyst and has semi-automaticity and advantages for complex models [35,38]. The SWAT-CUP software was used in this study to simulate the runoff of the Huangbizhuang Station under various land-use scenarios utilizing the SUFI−2 algorithm for sensitivity analysis, calibration, and verification.

The SWAT models were constructed for five different land-use scenarios in 1980, 1990, 2000, 2010, and 2020, respectively, and the runoff simulations were conducted for the five different land-use scenarios. As shown in Table 2, the various land-use scenarios were classified into warm-up, calibration, and validation periods.

**Table 2.** Classification of model time periods.

| Land-Use Period | Period | Warm-Up Period | Calibration Period | Validation Period |
|---|---|---|---|---|
| 1980 | 1960–1979 | 1960 | 1961–1970 | 1971–1979 |
| 1990 | 1980–1989 | 1980 | 1981–1985 | 1986–1989 |
| 2000 | 1990–1999 | 1990 | 1991–1995 | 1996–1999 |
| 2010 | 2000–2009 | 2000 | 2001–2005 | 2006–2009 |
| 2020 | 2010–2020 | 2010 | 2011–2015 | 2016–2020 |

The SWAT model was calibrated and validated using the correlation coefficient ($R^2$) and the Nash–Sutcliffe efficiency (*NSE*) coefficient [39] as the basis for evaluating the accuracy of model simulation [40,41]. $R^2$ indicates the consistency of the trend of natural and simulated data and *NSE* measures the fitting degree of natural, and simulated data, and their formulas are as follows:

$$R^2 = \left[ \frac{\sum_{i=1}^{T}\left(Q_{n,i} - \overline{Q_n}\right)\left(Q_{s,i} - \overline{Q_s}\right)}{\sum_{i=1}^{T}\left(Q_{n,i} - \overline{Q_n}\right)\sum_{i=1}^{T}\left(Q_{s,i} - \overline{Q_s}\right)} \right]^2 \tag{1}$$

$$NSE = 1 - \left[ \frac{\sum_{i=1}^{T}\left(Q_{n,i} - Q_{s,i}\right)}{\sum_{i=1}^{T}\left(Q_{n,i} - \overline{Q_n}\right)} \right]^2 \tag{2}$$

where $T$ is the length of the natural time series, $Q_n$ is the natural flow, $Q_{n,i}$ is the natural flow of the sequence, $\overline{Q_n}$ is the average flow of the natural sequence, $Q_s$ is the simulated value, $Q_{s,i}$ is the simulated flow of the sequence, and $\overline{Q_s}$ is the average flow of the simulated sequence.

The closer the $R^2$ and *NSE* are to 1, the closer the simulated value is to the natural value, and the better the simulation effect. On a monthly scale, the SWAT simulation accuracy reaches the standard when $R^2 > 0.6$ and $NSE > 0.5$.

### 3.4. Analysis of Parameter Sensitivity

The *t*-test method in SWAT-CUP was used to analyze the global sensitivity of natural and simulated runoff. The Huangbizhuang Station's natural runoff after the reduction

was used. Nine factors with the most significant impacts on runoff were selected based on the results of the parameter sensitivity analysis (Table 3) and the recommendations of Zuo [42] and Chang [43]. The model was calibrated and validated using the periods shown in Table 3. The t-statistic indicates the sensitivity of the parameter; the more sensitive the parameter, the higher the absolute value. Parameter sensitivity is represented by the *p* value, and the closer it is to 0, the more significant it is.

**Table 3.** Results of parameter sensitivity analysis of the Huangbizhuang Station.

| Parameter | 1960–1979 | | 1980–1989 | | 1990–1999 | | 2000–2009 | | 2010–2020 | |
|---|---|---|---|---|---|---|---|---|---|---|
| | t-st. | *p* Value | t-st. | *p* Value | t-st. | *p* Value | t-st. | *p* Value | t-st. | *p* Value |
| CN2 | −7.04 | 0.00 | −4.94 | 0.13 | −5.27 | 0.12 | −4.56 | 0.14 | 0.22 | 0.83 |
| GW_REVAP | −0.21 | 0.83 | 1.91 | 0.31 | 1.88 | 0.31 | 1.77 | 0.33 | 0.56 | 0.60 |
| SOL_AWC | 0.74 | 0.47 | 0.62 | 0.64 | 0.77 | 0.58 | 0.43 | 0.74 | 1.45 | 0.21 |
| SOL_K | −0.70 | 0.49 | −1.56 | 0.36 | −1.49 | 0.38 | −1.46 | 0.38 | 0.16 | 0.88 |
| ALPHA_BF | −1.02 | 0.32 | 0.18 | 0.89 | 0.13 | 0.92 | 0.19 | 0.88 | 0.43 | 0.68 |
| GW_DELAY | −0.80 | 0.43 | 0.75 | 0.59 | 0.86 | 0.55 | 0.69 | 0.62 | 0.75 | 0.49 |
| GWQMN | −0.30 | 0.77 | 1.17 | 0.45 | 1.27 | 0.42 | 1.11 | 0.47 | −0.76 | 0.48 |
| ESCO | −0.77 | 0.45 | −0.39 | 0.76 | −0.31 | 0.81 | −0.34 | 0.79 | −0.03 | 0.97 |
| CH_N2 | −0.94 | 0.36 | 0.84 | 0.56 | 0.77 | 0.58 | 0.82 | 0.56 | 0.47 | 0.66 |

Note: t-st.: t-statistic.

## 4. Results

### 4.1. Calibration and Validation of SWAT

The model was calibrated and verified under different land-use scenarios by comparing the runoff simulated by the SWAT model with the natural runoff. The parameter results (Table 4), $R^2$, and *NSE* during the calibration and verification periods (Table 5) were obtained. The $R^2$ values of the calibration and validation periods were >0.6, with most of them >0.7. *NSE* was above 0.5, and mostly >0.7. The constructed SWAT model in the upper reaches of the Hutuo River had high simulation accuracy and good applicability.

**Table 4.** Results of parameter calibration of the Huangbizhuang Station.

| Parameter | Parameter Range | | Land-Use Scenarios | | | | |
|---|---|---|---|---|---|---|---|
| | Lower Limit | Upper Limit | 1980 | 1990 | 2000 | 2010 | 2020 |
| CN2 | 35.00 | 98.00 | 43.90 | 80.75 | 29.66 | 87.00 | 67.98 |
| GW_REVAP | 0.02 | 0.20 | 0.02 | 0.02 | 0.02 | 0.02 | 0.02 |
| SOL_AWC(1) | 0.00 | 1.00 | 0.16 | 0.42 | 0.18 | 0.14 | 0.20 |
| SOL_K(1) | 0.00 | 2000.00 | 8.36 | 5.16 | 3.42 | 18.26 | 25.93 |
| ALPHA_BF | 0.00 | 1.00 | 1.18 | 1.02 | 0.07 | 0.16 | 0.80 |
| GW_DELAY | 0.00 | 500.00 | 343.65 | 450.59 | 278.09 | 299.97 | 412.85 |
| GWQMN | 0.00 | 5000.00 | 1.41 | 1.44 | 2.05 | 2.05 | 3.12 |
| ESCO | 0.00 | 1.00 | 0.05 | 0.04 | 0.60 | 0.55 | 0.17 |
| CH_N2 | −0.01 | 0.30 | 0.06 | 0.03 | 0.20 | 0.31 | 0.16 |

**Table 5.** Results of the evaluation of monthly flow simulation accuracy of the Huangbizhuang Station.

| Land-Use Scenarios | Calibration Period | | | Validation Period | | |
|---|---|---|---|---|---|---|
| | Period | $R^2$ | *NSE* | Period | $R^2$ | *NSE* |
| 1980 | 1961–1970 | 0.83 | 0.82 | 1971–1979 | 0.65 | 0.63 |
| 1990 | 1981–1985 | 0.89 | 0.88 | 1986–1989 | 0.64 | 0.54 |
| 2000 | 1991–1995 | 0.82 | 0.65 | 1996–1999 | 0.96 | 0.73 |
| 2010 | 2001–2005 | 0.64 | 0.62 | 2006–2009 | 0.66 | 0.64 |
| 2020 | 2011–2015 | 0.68 | 0.65 | 2016–2020 | 0.64 | 0.63 |

### 4.2. Analysis of Land-Use Change

The changes in land-use types over the five periods in the Hutuo River Basin above the Huangbizhuang Reservoir show that the rate of change in the area of arable land, forest land, and grassland in the basin was between −2.9% and 1.4% in all years, with decreases year-on-year (Table 6). The areas of arable land, forest land, and grassland decreased by 704 km$^2$ in total between 1980 and 2020, with a rate of change of −3.1%. The extent of urban development land expanded year-on-year, expanding 1.5 times between 1980 and 2020. The growth rate of urban construction land was less than 20% in all years until 2010, and it increased significantly between 2010 and 2020, with a growth rate of 72.6%.

**Table 6.** Land-use area and year-on-year changes in the Hutuo River Basin above the Huangbizhuang Station.

| Land-Use Type | Area of Land-Use Type in Each Decade (km$^2$) | | | | | Rate of Change of Land-Use Area by Decade (%) | | | | |
|---|---|---|---|---|---|---|---|---|---|---|
| | 1980 | 1990 | 2000 | 2010 | 2020 | 1980 | 1990 | 2000 | 2010 | 2020 |
| Arable land | 6282 | 6229 | 6316 | 6236 | 6113 | - | −0.8 | 1.4 | −1.3 | −2.0 |
| Forest land | 6932 | 6903 | 6841 | 6820 | 6718 | - | −0.4 | −0.9 | −0.3 | −1.5 |
| Grassland | 9769 | 9794 | 9765 | 9731 | 9445 | - | 0.3 | −0.3 | −0.4 | −2.9 |
| Water | 358 | 361 | 332 | 338 | 306 | - | 0.7 | −8.0 | 1.9 | −9.5 |
| Urban construction land | 528 | 581 | 636 | 761 | 1314 | - | 10.0 | 9.5 | 19.6 | 72.6 |
| Unused land | 62 | 63 | 42 | 45 | 37 | - | 1.6 | −34.4 | 9.2 | −18.2 |

### 4.3. Analysis of Results

Using the SWAT model established for the five land-use types, the daily runoff process from 1960 to 2020 under different land-use scenarios was simulated and compared with the natural runoff process for analysis, as shown in Figures 3–7. The simulated flood peak was compared with the natural flood peak and the simulated annual runoff with the annual natural runoff, and the degree of impacts of land-use changes on the flood peak and annual runoff was determined (Tables 7 and 8).

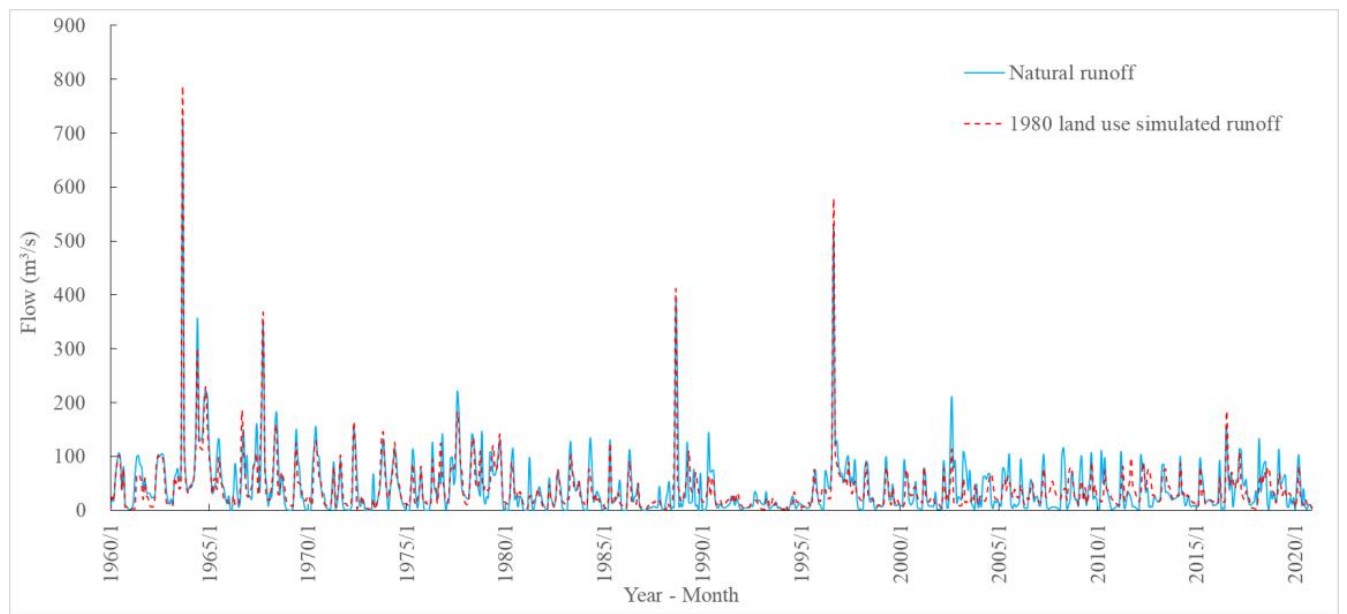

**Figure 3.** Comparison of simulated and natural runoff under the land-use scenario of 1980 at the Huangbizhuang Station.

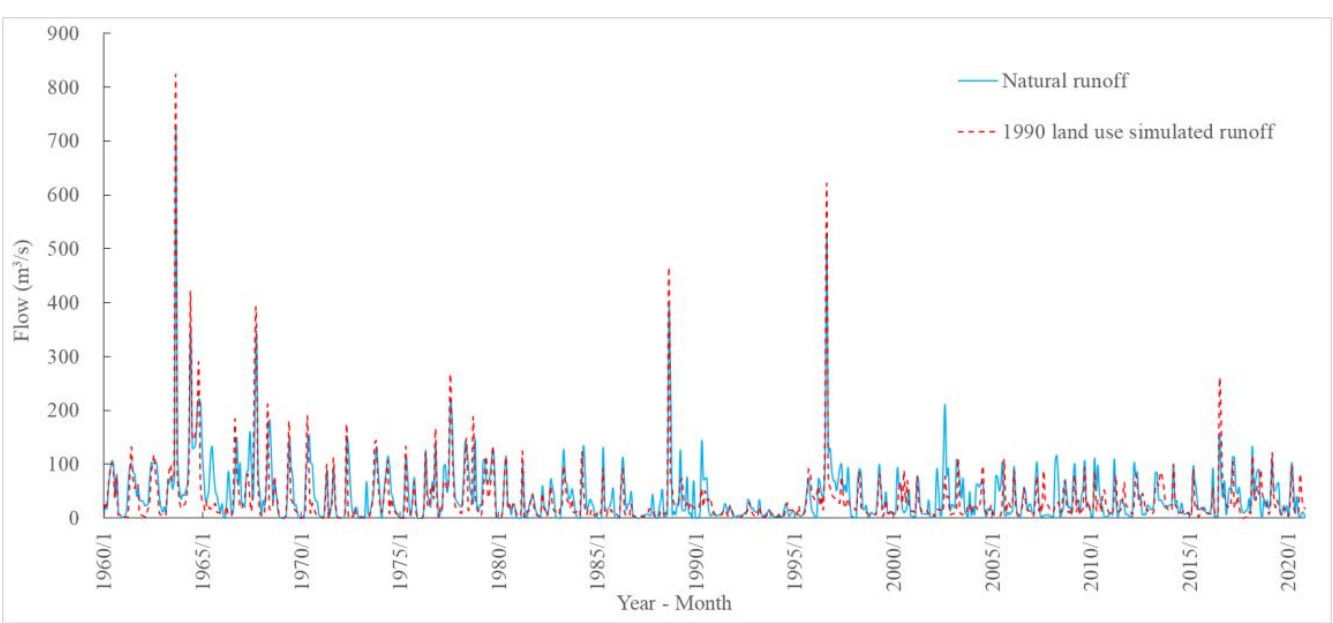

**Figure 4.** Comparison of simulated and natural runoff under the land-use scenario of 1990 at the Huangbizhuang Station.

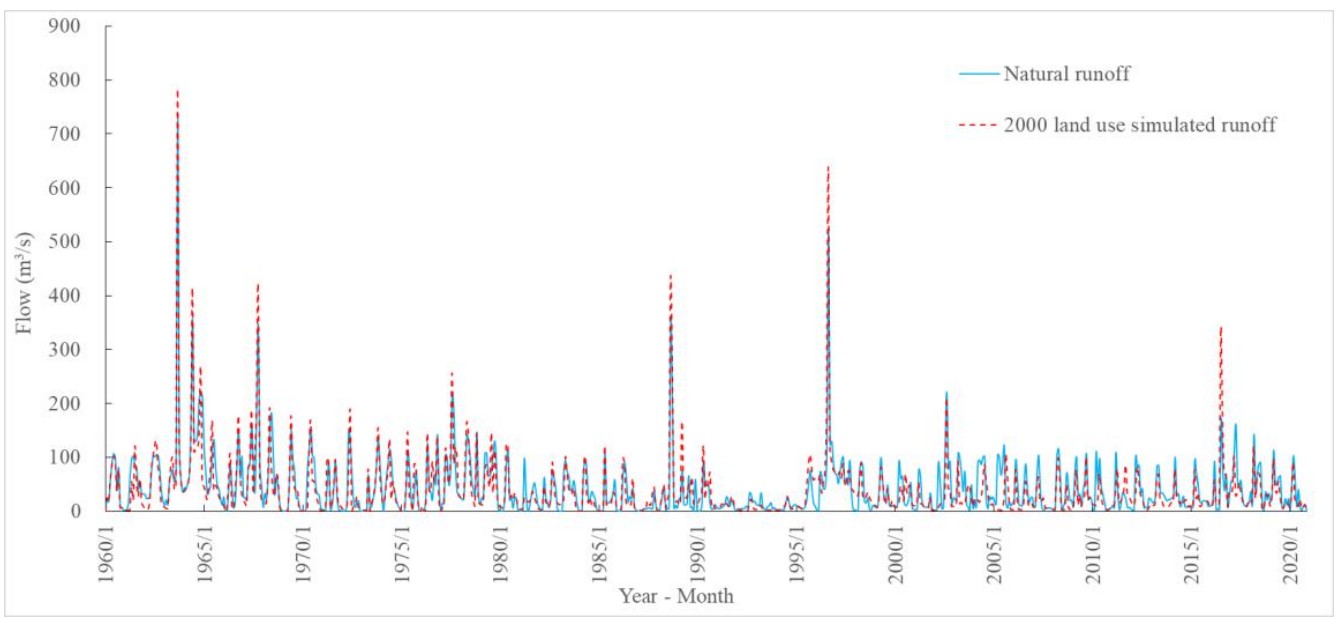

**Figure 5.** Comparison of simulated and natural runoff under the land-use scenario of 2000 at the Huangbizhuang Station.

### 4.3.1. Runoff Analysis under Different Land-Use Scenarios

According to the comparison and analysis of the simulated flood peak and simulated annual runoff and natural value in different periods under the five land-use scenarios in 1980, 1990, 2000, 2010, and 2020, it can be seen that under the same land-use scenario, with an increase in the simulation period, the error between the simulated value and the natural value gradually changes from positive to negative, and compared with the natural value, the simulated value showed a decreasing trend. For example, the flood peak and annual runoff from 1980 to 2020 for the land-use scenario in 1980 were compared with their natural values. The simulated flood peak and annual runoff were in good agreement with the natural values from 1960 to 1979 when compared to the natural runoff. The

difference between the simulated and natural flood peak was −5.7%, while the difference between the simulated and natural annual runoff was 1.6%. The differences between the simulation and natural values for the flood peak were 23.0, −11.7, −26.9, and 16.8% for 1980–1989, 1990–1999, 2000–2009, and 2010–2020, respectively. The differences between the simulation and natural values for annual runoff were 5.1, −3.6, −11.8, and 20.3% for 1980–1989, 1990–1999, 2000–2009, and 2010–2020, respectively. The simulated values of the flood peak and annual runoff were lower than those of natural values.

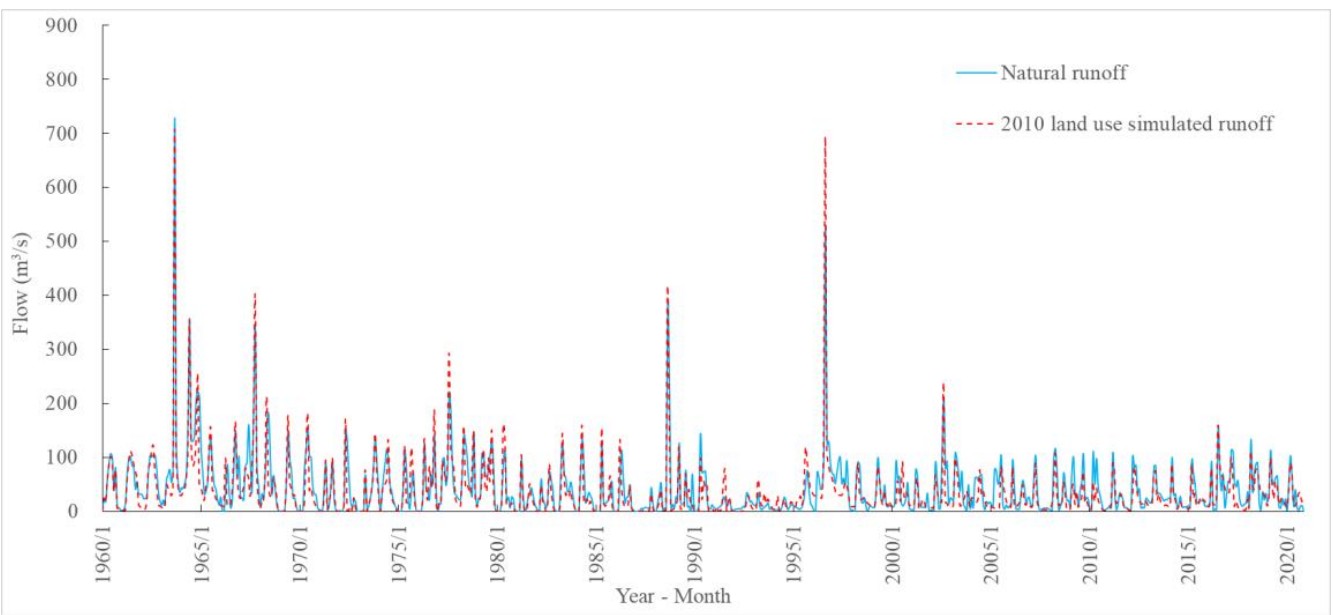

**Figure 6.** Comparison of simulated and natural runoff under the land-use scenario of 2010 at the Huangbizhuang Station.

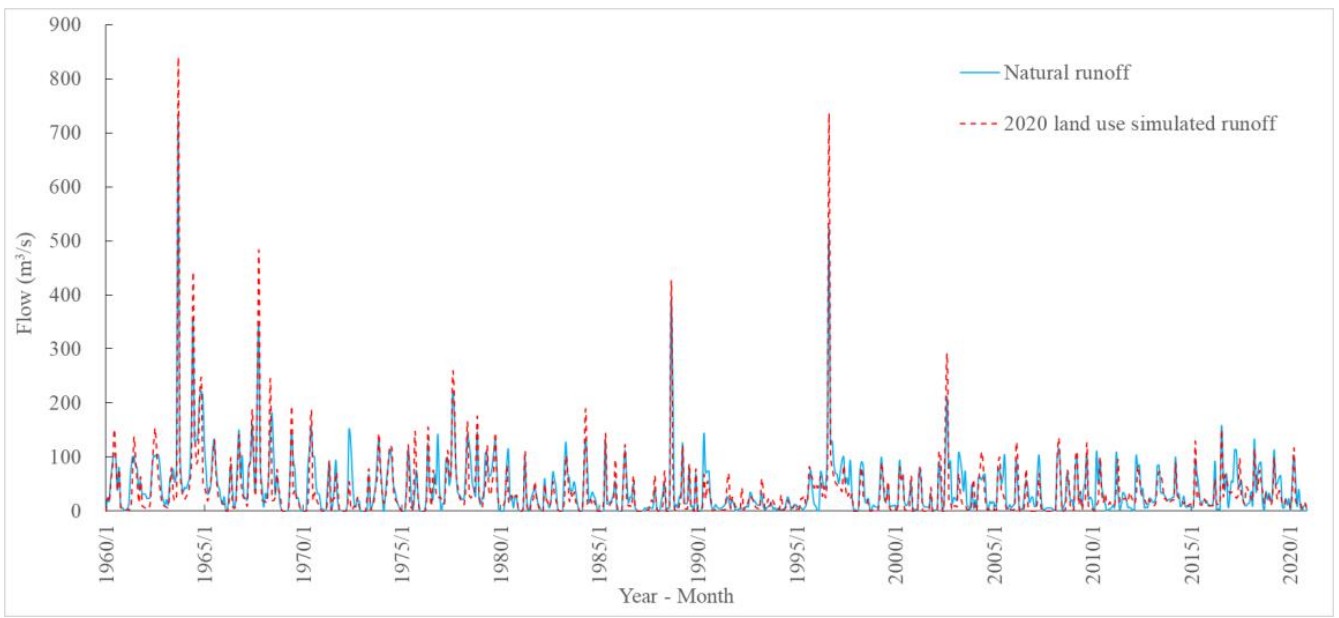

**Figure 7.** Comparison of simulated and natural runoff under the land-use scenario of 2020 at the Huangbizhuang Station.

**Table 7.** Comparative analysis of simulated and natural values of flood peaks under different land-use scenarios.

| Simulation Period | Natural Value (m³/s) | Simulated Values (m³/s) | | | | | Simulated Values—Natural Values (m³/s) | | | | | (Simulated Value—Natural Value)/Natural Value (%) | | | | |
|---|---|---|---|---|---|---|---|---|---|---|---|---|---|---|---|---|
| | | 1980 | 1990 | 2000 | 2010 | 2020 | 1980 | 1990 | 2000 | 2010 | 2020 | 1980 | 1990 | 2000 | 2010 | 2020 |
| 1960–1979 | 188.0 | 181.0 | 209.6 | 214.6 | 207.6 | 219.7 | −7.0 | 21.6 | 26.7 | 19.7 | 31.7 | −5.7 | 9.9 | 14.8 | 12.7 | 15.7 |
| 1980–1989 | 134.6 | 113.4 | 124.9 | 131.2 | 151.1 | 142.1 | −21.2 | −9.8 | −3.4 | 16.5 | 7.5 | −23.0 | −16.2 | −5.1 | 11.4 | 5.8 |
| 1990–1999 | 115.8 | 107.9 | 110.6 | 121.1 | 132.9 | 134.0 | −7.9 | −5.2 | 5.3 | 17.1 | 18.2 | −11.7 | −14.5 | −9.0 | 24.1 | 22.2 |
| 2000–2009 | 108.3 | 75.3 | 89.5 | 95.8 | 100.0 | 117.9 | −33.0 | −18.8 | −12.5 | −8.4 | 9.6 | −26.9 | −9.7 | −12.1 | −9.4 | 7.7 |
| 2010–2020 | 111.1 | 93.2 | 110.2 | 110.1 | 94.0 | 104.5 | −18.0 | −0.9 | −1.0 | −17.1 | −6.6 | −16.8 | −3.2 | −6.2 | −15.8 | −5.1 |
| 1960–2020 | 131.6 | 114.2 | 129.0 | 134.6 | 137.1 | 143.7 | −17.4 | −2.6 | 3.0 | 5.5 | 12.1 | −16.8 | −6.7 | −3.5 | 4.6 | 9.3 |

**Table 8.** Comparative analysis of simulated and natural values of annual runoff under different land-use scenarios.

| Simulation Period | Natural Value (m³/s) | Simulated Values (m³/s) | | | | | Simulated Values—Natural Values (m³/s) | | | | | (Simulated Value—Natural Value)/Natural Value (%) | | | | |
|---|---|---|---|---|---|---|---|---|---|---|---|---|---|---|---|---|
| | | 1980 | 1990 | 2000 | 2010 | 2020 | 1980 | 1990 | 2000 | 2010 | 2020 | 1980 | 1990 | 2000 | 2010 | 2020 |
| 1960–1979 | 60.9 | 61.9 | 67.9 | 65.9 | 68.9 | 69.9 | 1.0 | 7.0 | 5.0 | 8.0 | 9.0 | 1.6 | 11.5 | 8.2 | 13.1 | 14.8 |
| 1980–1989 | 31.5 | 29.9 | 29.5 | 31.1 | 35.5 | 36.5 | −1.6 | −2.0 | −0.5 | 4.0 | 5.0 | −5.1 | −6.3 | −1.5 | 12.7 | 15.8 |
| 1990–1999 | 31.8 | 30.7 | 28.1 | 34.6 | 32.8 | 33.8 | −1.1 | −3.7 | 2.8 | 1.0 | 2.0 | −3.6 | −11.7 | 8.7 | 3.1 | 6.3 |
| 2000–2009 | 35.7 | 31.5 | 29.1 | 29.8 | 32.7 | 41.7 | −4.2 | −6.6 | −6.0 | −3.0 | 6.0 | −11.8 | −18.5 | −16.7 | −8.4 | 16.8 |
| 2010–2020 | 34.5 | 27.5 | 31.2 | 29.3 | 29.5 | 33.5 | −7.0 | −3.3 | −5.1 | −5.0 | −1.0 | −20.3 | −9.5 | −14.9 | −14.5 | −2.9 |
| 1960–2020 | 38.9 | 36.3 | 37.2 | 38.1 | 39.9 | 43.1 | −2.6 | −1.7 | −0.8 | 1.0 | 4.2 | −6.7 | −4.4 | −2.0 | 2.6 | 10.8 |

When compared with other years, the land-use type in 1980 was the largest sum of cultivated land, forest land, and grassland, and the smallest area of urban construction land. At the same time, the simulated values for flood peak and annual runoff from 1960–1979 were also higher than those in other simulation periods. With the increase in simulation periods, the simulated values for flood peak and annual runoff also showed an increasing trend. These indicated that the increase in the area of urban construction land and the decrease in areas of forest land and grassland led to an increase in flood peak and annual runoff volume. The simulated flood peak and annual runoff under the remaining four land use scenarios, 1990, 2000, 2010, and 2020, followed the same law.

4.3.2. Analysis of Runoff Influenced by Land-Use Changes in Each Period

The impact of land use change on flood peak and annual runoff was analyzed from the five simulation periods of 1960–1979, 1980–1989, 1990–1999, 2000–2009, and 2010–2020. From 1960–2020, the differences between simulated and natural values for the flood peaks of the five land-use scenarios were −16.8, −6.7, −3.5, 4.6, and 9.3%, respectively, and the errors between the simulated and natural values of annual runoff were −6.7, −4.4, −2.0, −2.6, and 10.8%, respectively. The differences between the simulated and natural values for flood peak and annual runoff increased with the duration of land use.

For the five land-use scenarios of 1980, 1990, 2000, 2010, and 2020 within the 1960–1979 simulation period, the differences between the simulated and natural values for flood peaks and annual runoff were −5.7, 9.9, 14.8, 12.7, and 5.7% and 1.6, 11.5, 8.2, 13.1, and 14.8%, respectively. It can also be seen that the difference between the simulated and natural values for flood peak and annual runoff during the simulation period tended to increase with the duration of land use, and the simulated value under the land use scenario after 1980 is higher than the natural value. It showed that, in the same simulation period, the flood peak and annual runoff increased with the increase in urban construction land area and the decrease in cultivated land, forest land, and grassland area. The simulated flood peak and simulated annual runoff of the other four simulation periods, 1980–1989, 1990–1999, 2000–2009, and 2010–2020, also followed this law.

## 5. Discussion

The land-use characteristics of the upper Hutuo River Basin in 1980, 1990, 2000, 2010, and 2020 show that the areas of arable land, forest land, and grassland decreased year-on-year, with a total decrease of 704 km$^2$. Meanwhile, the area of urban construction land increased year-on-year, and in 2020, it was 1.5 times more than that in 1980. The differences between the simulated and natural values for the flood peaks and annual runoff in five land-use scenarios gradually increased from −16.8 to 9.3% and −6.7% to 10.8%, respectively. A comparison of land-use scenarios between 1960 and 2020 shows an increase in the area of urban construction land and a reduction in the areas of arable land, forest land, and grassland in the Hutuo River Basin, which led to an increase in flood peak and annual runoff volume. This result is compatible with the findings of Zhai [44] and Zhao [25]. According to Zhai [44], forestland in the Hutuo River Basin had the maximum water production, followed by grassland and farmland; however, the urban area had a lower flow yield than those of forestland, grassland, and cropland owing to the hardening of the substrate. Zhao [25] suggested that the restoration of vegetation in the Hutuo River Basin's Wutai Mountain area, following grazing restrictions and the restoration of cropland to forest, resulted in an increase in water resources. When compared to other land-use types, forest soils have a longer root system, a humus layer with higher water-holding properties, and a layer of dead branches and leaves, and therefore have a greater ability to absorb and use deep soil water and retain rainfall, resulting in a lower volume of runoff [45,46]. Due to its large soil pores and large saturated water content, grassland takes longer to fill all the pores in the soil with water and takes longer to produce runoff, delaying the production of flow and producing a smaller volume of water [47]. Arable land has low ground coverage; the water is very easy to infiltrate, and tillage frequently turns the soil, increasing soil porosity

and soil permeability, increasing soil water content and reducing runoff capacity [48]. In contrast, urban construction land is dominated by impermeable surfaces. When compared with permeable surfaces, such as forest land, arable land, and grassland, urban construction land has a small rainfall infiltration capacity, storage capacity, and water storage capacity, and a high flow-producing capacity [49]. As a result, the reduction in the area of woodland, arable land, and grassland reduces the leaf area index, the amount of water absorbed by the root system, the amount of water retained by the forest canopy, and the amount of transpiration by the vegetation. At the same time, the soil porosity becomes smaller, the infiltration rate of the soil becomes smaller, the proportion of midloam flow to the total runoff becomes smaller, the time of flow production becomes shorter, and the runoff process becomes steeper, as well as increases in peak flow and annual runoff [50–52]. The increase in construction land area leads to an increase in impervious area, land leveling, improved stormwater drainage systems, and a reduction in surface roughness, resulting in a reduction in the steady loss of rainfall, soil infiltration rate, subsurface runoff, and dry runoff, and an increase in annual runoff volume [53]. At the same time, urban storage capacity decreases, runoff confluence speeds up significantly, runoff gradients steepen, and peak flood flows increase [54]. In summary, as arable land, forest land, and grassland are transformed into urban construction land, the initial and steady loss of rainfall decreases, the infiltration rate of soil decreases, the hardening rate of the ground increases, the water holding capacity of the watershed decreases, the storage capacity of the basin decreases, and the annual runoff volume, as well as the peak flood flow, increases accordingly [55].

## 6. Conclusions and Suggestions

The SWAT model of the research region was built using hydrological and meteorological data, a digital elevation model, a soil-type map, and five land-use maps covering the upper reaches of the Hutuo River. After calibration and verification, the simulated runoff was better fitted to the natural runoff. The correlation coefficients between the model calibration and verification periods were >0.6, with the majority > 0.7. The Nash–Sutcliffe efficiency coefficient and root-mean-square error indicated the accuracy of the simulation. Our analyses and comparisons of the influences of different land-use scenarios over a long time series of runoff from 1960 to 2020 and land-use changes on runoff in each period showed the following:

(1)  The area of urban construction land in the research region gradually increased from 1980 to 1990, 2000, 2010, and 2020, with an overall increase of 1.5 times by 2020 when compared to 1980;

(2)  The areas of arable land, forest land, and grassland decreased gradually, with a total decrease of 704 km$^2$ and a change rate of −3.1%;

(3)  Flood peak and annual runoff volume increased by 25.8 and 18.7%, respectively.

Therefore, it can be concluded that the increase in the area of urban construction land and the decrease in areas of arable land, forest land, and grassland during 1960–2020 has increased over the impervious area, with a lower soil infiltration rate and lower underground and dry runoff in the upper Hutuo River Basin, which, in turn, has led to significant increases in flood peak and annual runoff.

An analysis of the effects of land-use change on flood processes revealed that an increase in the area used for urban construction would result in an increase in the annual runoff and flood peak in the upper reaches of the Hutuo River. However, an increase in the area used for forest land or grassland would result in an increase in the initial and stable losses, such as interception, depression filling, and infiltration, and decreases in flood peak and flood volume. Therefore, it is recommended that local government departments introduce the relevant policies for the protection of grasslands and forests, which are in the vicinity of the hillsides to facilitate reforestation and planting, restore and increase vegetation coverage, and enhance the capacity of the Hutuo River Basin to maintain a balance in water distribution, enabling it to withstand floods and droughts, and preventing soil erosion.

**Author Contributions:** Conceptualization, B.L. and J.Y.; methodology, B.L., J.Y. and J.S.; validation, J.S., J.Y., Y.L., X.Z. and R.L.; data curation, X.Z., R.L., J.Y. and Y.L.; writing—original draft preparation, J.Y. and B.L.; writing—review and editing, J.S. and B.L. All authors have read and agreed to the published version of the manuscript.

**Funding:** This research was funded by the National Natural Science Foundation of China (Grant no. 51879066).

**Institutional Review Board Statement:** Not applicable.

**Informed Consent Statement:** Not applicable.

**Data Availability Statement:** All data used during the study are proprietary or confidential and may only be provided with restrictions.

**Acknowledgments:** The authors are also thankful for the support from the Hebei University of Engineering.

**Conflicts of Interest:** The authors declare no conflict of interest.

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
