# Peer review of "Analysis of Runoff According to Land-Use Change in the Upper Hutuo River Basin"

_water, doi:10.3390/w15061138_

Round 1

Reviewer 1 Report

The paper presents an interesting and consistent approach to land use change and runoff applied to a watershed of social and economic importance, with a long historical database (an excellent amount of measured data is presented). Model calibration and validation steps are performed correctly. Although interesting, in a scientific context it is necessary to evidence the innovative aspects.

Three questions:

Could the error in classifying the type of use (Arable land, Forest land, Grassland) based on the use of remote sensing images interfere with the analysis presented in the paper?

Based on the values presented in table 6 and in the previous question, it is possible to infer that there are significant changes in land use (mainly for Arable land, Forest land, Grassland; percentage differences are low) or just evidence of an increase in urban areas (percentage high)?

“Flood peak and annual runoff volume increased by 25.8 and 18.7%.” How to identify the percentage of influence of the reduction in areas of Arable land, Forest land, Grassland and the increase in urban construction land on these Flood peak and annual runoff volume values?

Reviewer 2 Report

General comments

Liu et al. present a case study about the effects of land-use changes in on runoff in the Upper Hutuo River Basin using the SWAT model. The study is relevant for Water and merits publication therein.

In general, I think the work was well done. I cannot comment much on the specifics of the model setup since this is outside of my expertise. However, I think to make this paper ready for publishing a few things should be addressed: First of all, some sections of the paper are very long and I often do not see how it is relevant to the story the authors want to tell (see also specific comments). Especially the results section just lists results that are already presented in tables and figures. This should be shortened. On the other hand, the discussion of results remains very superficial and short. This part should be extended.

I recommend acceptance of the paper upon major revisions.

Specific comments

 Citation style in the text should be updated with brackets: [..]

Introduction                                                      

14-15 Is that statement really true? A quick search shows a lot of articles published in recent years addressing this very topic. Please explain.

55-106 This paragraph keeps adding disconnected statements Wang et al. did this…Tang et al used that. Could the authors please rewrite this in a more coherent manner. Most of these statements just briefly mention a study that was done, but not the outcome/relevance for this study (e.g. lines 84-85, 97-99, 99-102, 102-104, 105-106)

Overview of research area and methods

112 Heading should only be called "Overview of the research area" since "Methodology" follows in a separate paragraph

Methodology

154 "Table 1 displays the precision…" there is no display of precision in Table 1, just resolution of the datasets

Table 1 (1) Was the meteorological data a raster dataset? If so, what resolution. Could be mentioned here. (2) The "Accuracy/Scale" column should just be named "Scale" or better "Resolution" since there is no information on accuracy of the data.

166-168 Ambiguous phrase. Did the authors mean the amount of water abstracted upstream was added to the measured runoff? Please clarify.

Results

229-233 What was the method for reclassification? This part should be in the Methodology section

238 "…with a rate of change of -3.1%." A yearly rate?

253ff The entire paragraphs on "Runoff analysis under different land-use scenarios" and "Analysis of runoff influenced by land-use changes in each period" are extremely repetitive and abundant, since the results are already presented Tables 7 and 8 and Figures 3-7. As is, this is not very interesting to read. Please rewrite this by choosing the most relevant results and summarise them consicely. For the rest refer to the tables and figures provided.

Tables 7 & 8 are not referenced in the text. Please correct.

Figures 3-7 are not referenced in the text. Please correct.

Discussion

Overall, the discussion is very short and remains superficial. Please extend.

389 Discussion is a main heading ("4. Discussion")

Round 2

Reviewer 2 Report

General comments

Most comments from the first round were addressed appropriately in the revised manuscript. However, there are still some major issues that should be fixed before publication can be considered. This concerns mostly the Introduction and Discussion section.

The newly added paragraph (lines 77-99) still adds one sentence after the other without any coherence and storyline whatsoever connecting them. The only thing that was changed was that some more study results were mentioned. As a reader, I ask myself where the authors are going with statement after statement on different aspects of the effects of land-use changes on runoff.

The same goes for the changed paragraph (lines 122-141) on the specifics of runoff and land-use in the Hutuo River Basin. Although the different studies and their results are mentioned, the authors provide no synthesis towards their own research. There is only a disconnected short paragraph stating what was studied.

The Discussion section needs some attention as well. There are missing references and statements that cannot be made based on the authors works (see also minor comments below).

These issues should be addressed before publication can be considered.

Minor comments

Suggestion: Figures 3-7 should have the same scale on the y-axis to make comparisons between them easier. Maybe the figures could even be shown as only one figure. I'd leave that to the authors' discretion.

513-534 There are no supporting literature references in this paragraph. Please add. In general, there are a lot of definitive statements that cannot be supported by this study since this was not investigated (e.g. "Reducing cultivated land and grassland reduced soil porosity, making water less susceptible to infiltration and increasing annual runoff."). I ask the authors to be more diligent in what definitive claims they can make based on their own work. For everything else they need literature references.

643-644 I did not find this reference in any search. This goes for some other references (e.g. [27], [28], [29]) as well. Was the original language Chinese? If so, this should be made clear in the reference list.
